# CX3CL1 Action on Microglia Protects from Diet-Induced Obesity by Restoring POMC Neuronal Excitability and Melanocortin System Activity Impaired by High-Fat Diet Feeding

**DOI:** 10.3390/ijms23126380

**Published:** 2022-06-07

**Authors:** Jineta Banerjee, Mauricio D. Dorfman, Rachael Fasnacht, John D. Douglass, Alice C. Wyse-Jackson, Andres Barria, Joshua P. Thaler

**Affiliations:** 1Department of Medicine, Division of Metabolism, Endocrinology and Nutrition, University of Washington Medicine Diabetes Institute, University of Washington, Seattle, WA 98109, USA; jineta.banerjee@sagebase.org (J.B.); dorfmanm@uw.edu (M.D.D.); rachael.fasnacht@gmail.com (R.F.); jdaviddouglass@gmail.com (J.D.D.); wysejaca@gmail.com (A.C.W.-J.); 2Department of Physiology and Biophysics, University of Washington, Seattle, WA 98109, USA; barria@uw.edu

**Keywords:** microglia, fractalkine, POMC neurons, obesity, melanocortin system

## Abstract

Both hypothalamic microglial inflammation and melanocortin pathway dysfunction contribute to diet-induced obesity (DIO) pathogenesis. Previous studies involving models of altered microglial signaling demonstrate altered DIO susceptibility with corresponding POMC neuron cytological changes, suggesting a link between microglia and the melanocortin system. We addressed this hypothesis using the specific microglial silencing molecule, CX3CL1 (fractalkine), to determine whether reducing hypothalamic microglial activation can restore POMC/melanocortin signaling to protect against DIO. We performed metabolic analyses in high fat diet (HFD)-fed mice with targeted viral overexpression of CX3CL1 in the hypothalamus. Electrophysiologic recording in hypothalamic slices from POMC-MAPT-GFP mice was used to determine the effects of HFD feeding and microglial silencing via minocycline or CX3CL1 on GFP-labeled POMC neurons. Finally, mice with hypothalamic overexpression of CX3CL1 received central treatment with the melanocortin receptor antagonist SHU9119 to determine whether melanocortin signaling is required for the metabolic benefits of CX3CL1. Hypothalamic overexpression of CX3CL1 increased leptin sensitivity and POMC gene expression, while reducing weight gain in animals fed an HFD. In electrophysiological recordings from hypothalamic slice preparations, HFD feeding was associated with reduced POMC neuron excitability and increased amplitude of inhibitory postsynaptic currents. Microglial silencing using minocycline or CX3CL1 treatment reversed these HFD-induced changes in POMC neuron electrophysiologic properties. Correspondingly, blockade of melanocortin receptor signaling in vivo prevented both the acute and chronic reduction in food intake and body weight mediated by CX3CL1. Our results show that suppressing microglial activation during HFD feeding reduces DIO susceptibility via a mechanism involving increased POMC neuron excitability and melanocortin signaling.

## 1. Introduction

Obesity is a leading cause of morbidity and mortality worldwide with few effective treatments [1]. Current evidence suggests that diet-induced obesity (DIO) is associated with dysfunction of pro-opiomelanocortin (POMC) neurons, a hypothalamic cell population that maintains energy homeostasis by modulating food intake and energy expenditure [2,3,4]. POMC neurons are activated by nutrients and hormones, such as the adipokine leptin, reducing food intake through synaptic release of neurotransmitters (mostly glutamate) and alpha-melanocyte stimulating hormone (α-MSH), a neuropeptide melanocortin 3/4 receptor (MC3/4R) agonist [5,6,7]. Genetic defects in the leptin–melanocortin pathway such as mutations in MC4R lead to profound high-fat diet (HFD) sensitivity and increased obesity susceptibility in rodents and humans [2,8,9,10,11,12]. Likewise, nongenetic POMC neuron defects induced by HFD exposure, including alterations to α-MSH production, synaptic organization, and intracellular calcium responses, likely contribute to DIO pathogenesis [3,4,13,14].

The factors that drive neuronal dysfunction in obesity have not been fully identified, but growing evidence implicates hypothalamic inflammation as an important contributor [15,16,17,18,19,20,21]. Activation of microglia (CNS macrophage-like immune cells) and elevated expression of inflammatory cytokines are detectable in the mediobasal hypothalamus (MBH) within the first week of HFD exposure [15,16,17,22], even before inflammation is detected in peripheral metabolic organs [16]. Preventing gliosis through downregulation of microglial NF-κB signaling or ablating microglia altogether restores leptin sensitivity in hypothalamic neurons in vivo and leads to protection from HFD-associated weight gain [19,20]. Though the precise mechanisms remain to be determined, several recent studies connect these microglial-mediated metabolic benefits with the melanocortin system. In mice subjected to HFD or other inflammatory stimuli, microglia become closely apposed to the surface of POMC neuron cell bodies, suggesting an increase in intercellular signaling [21,23]. Moreover, DIO mouse models with genetic manipulations to either enhance or suppress microglial activation show corresponding alterations in POMC neuron cytoarchitecture, c-Fos positivity, and cell survival [21,24,25,26]. Though provocative, these studies leave undetermined the direct effect of microglia on POMC neuron electrophysiological properties and the requirement for melanocortin signaling in the regulation of DIO susceptibility by microglia.

In the CNS, the chemokine CX3CL1 (also known as fractalkine) is produced by neurons and binds a unique receptor CX3CR1 exclusively expressed by microglia, playing an important role to maintain microglial quiescence [27]. Full-length CX3CL1 is a membrane-bound molecule that can signal bidirectionally [28,29], but a soluble ligand consisting of the N-terminal signaling domain is released extracellularly by proteolytic processing. While there are some contexts in which differences between the functions of the membrane-bound and secreted CX3CL1 isoforms have been observed, the overall effect of reduced CX3CL1–CX3CR1 signaling (e.g., in homozygous CX3CR1*^gfp^* knock-in mice) is excessive microglial activation, leading to increased susceptibility to CNS inflammatory and neurodegenerative pathologies [30,31,32,33]. Similarly, MBH CX3CL1–CX3CR1 signaling decreases during HFD exposure, suggesting a potential mechanism by which HFD feeding triggers microglial activation and weight gain [20]. Indeed, restoring CX3CL1 expression in the MBH protects from HFD-induced obesity (DIO) [20], but the mechanisms linking CX3CL1 action—and more generally, microglial activation—to metabolic alterations remain unclear. Here, we provide evidence that inhibition of HFD-induced microglial activation by CX3CL1 restores POMC neuron excitability and correspondingly reduces weight gain in a melanocortin-dependent manner.

## 2. Results

### 2.1. Hypothalamic CX3CL1 Reduces Food Intake and Body Weight and Improves Leptin Sensitivity in HFD-Fed Mice

Our previous study demonstrated that DIO is associated with downregulation of CX3CL1–CX3CR1 signaling in the MBH of male mice, and that central CX3CL1 protects from diet-induced microglial inflammation and weight gain [20]. To determine how microglial–neuronal interactions are involved in these effects, we sought to avoid confounding by potential outside-in signaling in neurons through the intercellular domain of CX3CL1. Therefore, we first verified that MBH overexpression of the soluble ligand form, achieved by bilateral injections of AAVs containing CX3CL1-s (Figure 1A–C), was sufficient to have metabolic benefits. Two weeks after recovering from stereotaxic surgery, mice were placed on an HFD, and body weight and food intake were recorded. Reduction in weight gain and food intake over 25 days of HFD feeding was observed in AAV–CX3CL1-s when compared with the AAV–GFP control (Figure 1D–E). AAV–CX3CL1-s reduced food intake during the first week of HFD (Figure 1E), specifically between days 3 and 7 (Figure 1F). These data show that CX3CL1-s expression in the hypothalamus is sufficient to recapitulate the anti-obesity effects of full length CX3CL1 [20] but ensures a microglial site of action since CX3CR1 is expressed exclusively by microglia in the CNS (Appendix A).

We previously demonstrated that central CX3CL1 administration can suppress microglial activation [20]. Since HFD-induced gliosis and inflammation are associated with leptin resistance [17,19], we hypothesized that hypothalamic CX3CL1-s overexpression would increase leptin sensitivity. Indeed, a once-daily injection with a moderate dose of leptin (2 µg/g i.p.) for 2 consecutive days starting on day 25 of HFD reduced food intake in AAV–CX3CL1-s mice but not AAV–GFP controls (Figure 1G; trend by day 1 and significant at day 2). Consistent with its ability to improve leptin sensitivity, AAV–CX3CL1-s increased hypothalamic expression of POMC in weight-matched AAV–GFP controls (Figure 1H).

### 2.2. Inhibition of Activated Microglia Restores Intrinsic Excitability of POMC Neurons in HFD-Fed Mice

Chronic HFD consumption is associated with hypothalamic microglial activation and alterations in the electrophysiological properties of POMC neurons [4,16,17,19,21,22,34], but a direct connection between these diet-induced responses has not been established. Whole-cell recordings of GFP-labeled POMC neurons in age-matched POMC-MAPT-GFP mice (Figure 2A–C) showed that exposure to HFD for 4 weeks did not alter resting membrane potential (Figure 2D; body weight: Chow 27.65 g ± 0.74 g, N = 11; HFD 33.63 g ± 1.13 g, N = 13; *p* = 0.003). A similar proportion of POMC neurons were found to be spontaneously active in both chow and HFD-fed mice (~28%; Figure 2E), with equivalent firing rates (Figure 2F). This suggests that neither HFD exposure nor GFP expression resulted in fundamental cellular abnormalities in POMC neurons. 

Previous studies suggest that POMC neurons act as “synaptic integrators” summating electrical inputs over prolonged timescales [35]. Thus, a time ramp with steps of extended durations can be used to elicit this type of intrinsic excitability and determine how it is affected by different environmental exposures. To perform the time ramp study, we examined the responses of POMC neurons evoked by a minimal suprathreshold stimulus of variable duration (100–1000 ms, 30 pA) in brain slices incubated with blockers of glutamatergic and GABAergic postsynaptic currents (a method previously validated in [21]). POMC neurons of HFD-fed mice and chow-fed mice responded with similar numbers of action potentials when stimulated by a short duration stimulus (Figure 2G,I; 100 ms, 30 pA; Chow: 1.5 ± 0.3, n = 14, N = 3, HFD: 1.2 ± 0.26, n = 17, N = 4). However, a longer duration stimulus of the same amplitude evoked fewer action potentials in POMC neurons from HFD-fed mice compared with those from chow-fed controls (Figure 2G,I, 1000 ms, 30 pA, Chow: 12.21 ± 2.19, n = 14, N = 3, HFD: 7 ± 1.0, n = 17, N = 4, adjusted *p*-value (diet) = 1.71 × 10^−9^, two-way ANOVA, Tukey’s HSD multiple comparison test). Importantly, this diminished response occurred without depolarization block and in the absence of inhibitory and excitatory inputs, suggesting that HFD exposure significantly lowered the intrinsic excitability of the POMC neurons.

Incubating the hypothalamic slices from HFD-fed animals for 2 h in 100 µM minocycline (a tetracycline derivative that inhibits microglial activation) [36,37,38] restored the number of evoked action potentials in HFD POMC neurons to chow levels (Figure 2H,J, Chow: 10.09 ± 1.36, n = 11, N = 3, HFD: 9.68 ± 1.37, n = 19, N = 3, adjusted *p*-value: HFD + Minocycline: Chow + Minocycline = 0.59; HFD + Vehicle: HFD + Minocycline = 0.0017, Interaction: Minocycline:diet Pr = 0.00137, three-way ANOVA, Tukey’s HSD multiple comparisons test), suggesting that microglial activation induced by HFD feeding is necessary for altering the intrinsic excitability of POMC neurons.

### 2.3. CX3CL1 Restores Intrinsic Excitability of POMC Neurons in HFD-Fed Mice 

Based on the minocycline data (Figure 2G–J), we hypothesized that the more specific microglial inhibitor CX3CL1 would restore POMC neuron function during HFD feeding, which might account for its anti-obesity properties (Figure 1). Hypothalamic slices from chow and HFD-fed POMC-MAPT-GFP mice were incubated with recombinant mouse CX3CL1 (20 nM; soluble form) for 2 h prior to recording. Similar to the effect observed with minocycline (Figure 2J), CX3CL1 restored the number of evoked action potentials in HFD–POMC neurons to chow levels (Figure 3A,B, 1000 ms, 30 pA, Chow + CX3CL1: 11.91 ± 2.36, n = 12, N = 3; HFD + CX3CL1: 11.27 ± 1.47, n = 11, N = 3; HFD + Veh: 6.70 ± 1.43, n = 10, N = 3, adjusted *p*-value: HFD + Vehicle: HFD + CX3CL1 = 0.0002; HFD + CX3CL1: Chow + CX3CL1 = 0.64; HFD + CX3CL1: Chow + Vehicle = 0.37, Intervention: CX3CL1:diet Pr = 1.56 × 10^−3^, three-way ANOVA, Tukey*’*s HSD multiple comparisons test). 

Next, we assessed synaptic inputs to POMC neurons. Spontaneous excitatory postsynaptic currents (sEPSCs) received by POMC neurons were unchanged by HFD exposure both in terms of amplitude and frequency (Figure 3C–E) so we did not test the effect of CX3CL1 on sEPSCs. In contrast, we observed a significant increase in the cumulative probability of larger amplitude spontaneous inhibitory postsynaptic currents (sIPSCs) in POMC neurons from HFD-fed mice vs. chow-fed (Figure 3F,G, red trace, chow vs. HFD: adjusted *p* value = 3.18 × 10^-11^, KS test, Benjamini–Hochberg multiple comparison correction) with no differences in frequency (Figure 3H). CX3CL1 application reduced the sIPSC amplitude in the HFD group toward chow levels (Figure 3G, blue trace, HFD + veh vs. HFD + CX3CL1: adjusted p value = 0.0019, KS test, Benjamini–Hochberg multiple comparison correction) though it also mildly increased sIPSC amplitude in chow subjects (Figure 3G, grey trace, Chow + veh vs. Chow + CX3CL1: adjusted *p* value = 0.0004, KS test, Benjamini–Hochberg multiple comparison correction). Notably, both CX3CL1-treated groups had significantly lower sIPSC amplitude than the HFD group, suggesting an overall restorative effect of CX3CL1 on the changes affecting POMC neurons. Together these results support the hypothesis that microglial CX3CL1–CX3CR1 signaling reverses the impairment of hypothalamic POMC neuronal activity induced by HFD feeding.

### 2.4. The Anti-Obesity Effect of Central CX3CL1 Requires Intact Melanocortin Signaling In Vivo

The degree of POMC neuron activity generally correlates with production of α-MSH, a neuropeptide derived from POMC that reduces food intake and body weight by acting on MC3R and MC4R [39,40]. Thus, based on the ability of CX3CL1 to restore POMC neuron excitability (Figure 3) and increase *Pomc* gene expression in vivo (Figure 1E), we hypothesized that CX3CL1 reduces food intake and body weight during DIO via increased melanocortin signaling. To test this hypothesis, we performed pharmacological blockade of the central melanocortin signaling in HFD-fed mice treated with CX3CL1-s. A single i.c.v. administration of recombinant mouse CX3CL1 peptide (soluble form) tended to reduce 24-h HFD intake (Figure 4B, Sal/Sal vs. Sal/CX3CL1) and significantly limited weight gain (Figure 4C, Sal/Sal vs. Sal/CX3CL1). Pre-treatment with a dose of the MC3R/MC4R antagonist SHU9119 (0.1 nmol, i.c.v.) that by itself had no effect on food intake and body weight (Figure 4B,C, Shu/Sal vs. Sal/Sal) blocked the metabolic benefits of acute CX3CL1 treatment (Figure 4B,C, Sal/CX3CL1 vs. Shu/CX3CL1). To assess the involvement of melanocortin system activation in DIO prevention by CX3CL1, s.c. osmotic minipumps containing SHU9119 (0.2 nmol/h) were placed in lateral ventricle-cannulated mice with prior AAV–CX3CL1-s or AAV–GFP injection into the MBH (Figure 4D). As described above, hypothalamic overexpression of CX3CL1-s reduced food intake and body weight in DIO mice (Figure 1A–C). However, concurrent SHU9119 infusion abolished these metabolic benefits (Figure 4E,F) despite an overall increase in body weight and food intake in both groups relative to untreated animals (compare Figure 4E,F to Figure 1A–C). Collectively, these data support a requirement for intact melanocortin signaling in the anti-obesity effect of microglial silencing via hypothalamic CX3CL1 overexpression.

## 3. Discussion

Neuronal output dictates behavior including food intake and metabolic rate, yet diet-induced activation of glial cells can promote susceptibility to hyperphagia and obesity [3,16,18,19,24,25]. Therefore, glial cells must influence energy balance through action on neighboring neurons. Here, we focus on one such glial–neuronal interaction via CX3CL1–CX3CR1 signaling to illustrate how microglial activation alters melanocortin system activity during HFD feeding to promote DIO. 

CX3CL1–CX3CR1 signaling maintains microglia in their basal surveillance mode with downstream impacts on neuronal function and survival in the adult brain [41]. CX3CR1 is a Gi protein-coupled receptor that regulates apoptotic, proliferative, transcriptional, and migratory functions in microglia [27]. Perturbation of CX3CL1–CX3CR1 signaling is associated with increased neuroinflammation and neuronal cell death in animal models of neurodegenerative disease [32,42,43,44,45]. In the context of DIO, where microglial inflammation is required for body weight gain upon HFD exposure [19,24,25], we have previously shown that CX3CL1 is reduced in the hypothalamus of male mice, and that central administration of CX3CL1 prevented diet-induced weight gain and microglial activation [20]. In this study, we investigated a potential mechanism by which this chemokine provides protection from weight gain. Given potential confounding by different signaling functions for the membrane-bound and soluble isoforms of CX3CL1 [28,29,42], we first determined that the soluble form is equally effective in preventing weight gain. This occurred despite the presence of endogenous CX3CL1 (at low levels due to HFD feeding [20]), suggesting that CX3CL1-s overexpression could have pharmacological benefits for obesity treatment. 

Mice overexpressing CX3CL1-s in the MBH displayed a significant reduction in food intake and an increased anorexigenic response to leptin, suggesting that microglial CX3CR1 signaling may affect neurons involved in feeding regulation. Indeed, we found that CX3CL1 increases the expression of the neuropeptide POMC in vivo. In support of this cell–cell interaction model, a previous study demonstrated an increase in microglia-POMC neuron cell contact when mice were exposed to a high-fat, high-sucrose diet [21]. The specificity of CX3CL1 action on microglia is further supported by our immunohistological analysis of Cx3cr1-Gfp mice confirming that CX3CR1 is not expressed in neurons, astrocytes, or tanycytes. Similarly, single-cell RNAseq analysis of mouse hypothalamic tissue has shown that CX3CR1 is not found in any CNS cell populations other than microglia and perivascular macrophages [46]. CX3CR1 is also expressed in subsets of peripheral monocytes and macrophages [47] so it is possible that circulating myeloid cells that infiltrate the MBH in HFD-fed mice [19] may also contribute to the anti-obesity effect of CX3CL1.

CX3CL1 acts as an anti-inflammatory molecule by downregulating microglial IL-1β, TNF-α and, IL-6 production [20,48,49]. We recently showed that central CX3CL1 administration to HFD-fed mice reduces the expression of TNFα in hypothalamic microglia [20]. There is accumulating evidence that TNFα is an important pro-obesogenic factor. Knockdown of TNFα signaling in MBH neurons of DIO mice reduces food intake and body weight gain, while lowering mitochondrial activity and cellular stress in POMC neurons [21]. Thus, we speculate that hypothalamic CX3CL1 may restore POMC neuron excitability and limit weight gain in DIO mice by reducing TNFα signaling. However, proinflammatory agents like TLR2 and TLR4 agonists also induce microglial activation but increase POMC neuron activity [23,38]. Thus, an important priority for future research is to distinguish the molecular characteristics of microglia activated by different inflammatory stimuli, such as HFD and LPS, in order to determine how these interventions elicit distinct metabolic outcomes. It is also important to consider an alternative to the reduced TNF signaling model, which posits that mitochondrial metabolism and reactive oxygen species production in microglia affect neighboring neurons, such as closely apposed POMC cells [21,25]. In support of this hypothesis, microglia-specific deletion of UCP2, a critical protein in mitochondrial function, prevents HFD-induced obesity and induces cytological markers of increased POMC neuronal activity [25]. Here, we have extended these important findings by demonstrating that microglial silencing ex vivo increases POMC neuron excitability and in vivo acts through the melanocortin system to reduce DIO susceptibility.

Hypothalamic inflammation and gliosis are proposed to mediate the synaptic input rearrangement, c-Fos induction, and neuronal loss observed in POMC neurons during HFD feeding [3,16,21]. Consistent with previous studies, we found that mice exposed to HFD have reduced POMC neuron activity due to alterations in intrinsic excitability, as well as changes in synaptic inputs [4]. Minocycline treatment restored neuronal excitability, suggesting that microglial activation contributes to the effect of HFD to inhibit POMC neurons. However, these data should be interpreted with caution as minocycline has been reported to affect other cell types [50]. Nevertheless, restoration of intrinsic excitability of POMC neurons in HFD-fed mice by CX3CL1 support a role for microglia in POMC neuron activity during DIO. Though CX3CL1–CX3CR1 signaling is known to modulate neuronal activity in other contexts, the mechanisms involved are not completely understood [51]. One possibility is suggested by a recent study demonstrating that reduced mitochondrial Ca^2+^ uptake during HFD feeding contributes to impaired POMC neuron excitability [4]. Activation of CX3CR1 signaling may alter the microglial secretome to restore POMC neuron activity through improved mitochondrial Ca^+2^ dynamics.

In our findings, CX3CL1 treatment of HFD-fed mice improved the anorexigenic response to leptin, restored POMC neuron excitability, and increased *Pomc* expression, all actions that would be predicted to raise the level of the melanocortin agonist, α-MSH, which acts at the MC4R to reduce food intake [52]. Indeed, both acute and chronic administration of SHU9119, an MC3/4 receptor antagonist, eliminated the protective effect of CX3CL1 on diet-induced body weight gain and food intake. Though the acute effect of icv-injected CX3CL1 may involve extrahypothalamic regions, including brain areas controlling food reward and motivation (important factors in the first 24 h of HFD feeding [53]), the administration of a subthreshold dose of SHU9119 still blocked the acute benefits of CX3CL1. Similarly, chronic MC4R antagonism eliminated the body weight differences between CX3CL1-overexpressing and control mice during DIO, indicating the metabolic benefits of CX3CL1—and by extension microglial silencing—are melanocortin signaling-dependent. Similar to previous studies using pharmacologic or genetic blockade of MC4R, SHU9119 infusion induced heavier animals across all categories (a likely result of chemogenetic, optogenetic, or ablation-based strategies to silence POMC neurons also) [54,55,56,57,58]. Nevertheless, studies that identified melanocortin-independent processes observed relative weight differences between treatment groups (e.g., activity-based anorexia [59,60], AgRP neuron ablation-induced anorexia [61], xenin-induced anorexia [62], histamine-induced anorexia [63], and cannabinoid-induced anorexia [64]). Thus, the failure of CX3CL1 to impact metabolic parameters with either acute or chronic MC4R antagonism indicates that restoring melanocortin signaling is a key downstream mechanism by which microglial inhibition reduces DIO susceptibility.

In summary, we demonstrate that CX3CL1—specifically its secreted soluble isoform—potently reduces weight gain when overexpressed in the MBH. CX3CL1 reverses HFD-induced alterations in POMC neuron excitability and inhibitory inputs. Moreover, the anti-obesity effect of CX3CL1 is blocked in the presence of a melanocortin receptor 3/4 antagonist. Together, these findings delineate a connection between microglial activation and POMC neuron dysfunction that contributes to DIO pathogenesis and demonstrate a potential metabolic benefit of strategies to target microglial inhibition.

## 4. Materials and Methods

### 4.1. Animals

Adult male C57BL/6J and Cx3cr1-GFP [47] mice, obtained from Jackson Laboratory (JAX stock #000664, #005582), and POMC-MAPT-GFP [65], a kind gift from Dr. Tamas Horvath (Yale University, New Haven, CT, USA), were housed in temperature-controlled rooms with 12:12 h light:dark cycle under specific pathogen-free conditions. The POMC-MAPT-GFP colony was maintained as homozygotes in our animal facility and used for electrophysiological measurement of POMC neurons. All procedures were performed in accordance with NIH Guidelines for Care and Use of Animals and were approved by the Animal Care Committee at the University of Washington.

### 4.2. Reagents

A plasmid containing the hybrid CMV-chicken β-actin promoter driving mRNA transcription of the soluble form of CX3CL1 (CX3CL1-s; aa 1-336) with a hemagglutinin (HA)-tag appended to the C-terminus [42] (kindly provided by Dr. Kevin Nash, University of South Florida, Tampa, FL, USA) was packaged into an AAV9 vector by the University of Washington Diabetes Research Center Viral Vector and Transgenic Mouse Core. The AAV9–GFP control virus was kindly provided by Dr. Michael W. Schwartz, University of Washington, Seattle, WA, USA. Carrier-free recombinant mouse CX3CL1 was used for intracerebroventricular (i.c.v.) administration (BioLegend, San Diego, CA, USA; Cat#583502), while CX3CL1 + BSA (carrier protein to enhance protein stability) was used for in vitro studies (R&D Systems, Minneapolis, MN, USA; 458-MF-025). Leptin was obtained from the National Hormone and Peptide Program (Dr. A. F. Parlow, Harbor-UCLA Medical Center Torrance, CA, USA) and SHU9119 from Bachem (Torrance, CA, USA; Cat#4027601).

### 4.3. Surgical Procedures

#### 4.3.1. CX3CL1-s Hypothalamic Overexpression

Two groups of 8- to 10-week-old male wild-type C57BL6/J mice were injected with AAV9–GFP or AAV9–CX3CL1-s (viral titer: 1 × 10^12^ viral particles/mL; n = 7/group). Briefly, mice from each AAV group underwent intrahypothalamic viral injections under isoflurane anesthesia. They received two consecutive stereotaxic injections (0.2 µL) bilaterally into the arcuate and ventromedial hypothalamus (anterior-posterior, −1.4 mm; lateral, ±0.5 mm; dorsal-ventral, −5.3 and −5.7 mm) using a Hamilton syringe (Cat#80030) with a 33-gauge needle at a rate of 50 nl/min (Micro4 controller, WPI. FL, USA) followed by a 7 min waiting period before needle removal. Mice were given 2 weeks to recover and acclimated to handling for 1 week before switching to an HFD (60% kcal fat; D12492; Research Diets, New Brunswick, NJ, USA). Body weight and food intake were recorded daily during the first week on the HFD and then twice weekly until the end of the study (25 days after diet switch). A leptin-sensitivity study was performed in these animals at the end of the study (see below). The location of viral infection/protein expression was assessed immunohistochemically, and, based on this postmortem evaluation, data from one mouse in the GFP control group that displayed no expression in the MBH were excluded from the final analysis.

#### 4.3.2. Acute SHU9119/CX3CL1 i.c.v. Injections

C57BL6/J male mice underwent implantation of an acute injection lateral ventricle (LV) cannula (Plastic One, Roanoke, VA, USA; co-ordinates: 0.7 mm posterior to bregma; 1.3 mm lateral, and 1.3 mm below the skull surface) under isoflurane anesthesia. Mice were given 2 weeks for recovery before i.c.v. administration. On the day of the experiment, four groups: (1) Saline/Saline, (2) SHU9119/Saline, (3) Saline/CX3CL1, (4) SHU9119/CX3CL1) of weight-matched mice were established (n = 5 per group). Food was removed 4 h prior to a first i.c.v. injection with saline or SHU9119 (0.5 nmol in 1µL) followed by a second i.c.v. injection of saline or CX3CL1 (1 µg in 1µL) 30 min later (See time scale in Figure 4A). Animals were then provided HFD, and food intake and body weight were measured after 24 h.

#### 4.3.3. Chronic SHU9119 i.c.v. Infusion

Two groups of 8- to 10-week-old male wild-type C57BL6/J mice were injected with AAV9–GFP or AAV9–CX3CL1-s as described above. During the same surgical procedure, mice underwent implantation of a chronic infusion LV cannula (Brain Infusion Kit 3, ALZET, Cupertino, CA, USA; Cat#0008851; coordinates: 0.7 mm posterior to bregma; 1.3 mm lateral, and 2.1 mm below the skull surface) under isoflurane anesthesia. A catheter tube filled with sterile saline was connected to the cannula and implanted subcutaneously in the scapular region of the animal. After 10 days of recovery time from surgery, daily measurements of body weight and food intake were recorded. Two days before switching to HFD, an osmotic minipump (model 1002, Alzet, Cupertino, CA, USA) containing SHU9119 was connected to the catheter and implanted subcutaneously for 14-day i.c.v. infusion (0.2 nmol/h at 0.25 µL/h). The rate of SHU9119 infusion was based on previous studies in rodents showing that this dose effectively blocks MC4R activity [66,67]. Measurements of body weight and food intake were continued for 3 weeks. The location of viral infection/protein expression was verified postmortem in all mice by IHC.

### 4.4. Leptin Sensitivity

Following a within-subject design, mice (AAV9–GFP and AAV9–CX3CL1-s; n = 7/group) fed with HFD for 28 days and individually housed received daily injections of saline (200 µL/day i.p.) for 2 days followed by daily recombinant murine leptin (2 µg/g body weight i.p.) for 2 days with body weight and food intake measured daily [68,69]. Injections were performed at 10 am and mice were maintained with ad libitum food access during the experiment.

### 4.5. Brain Slice Preparation and Electrophysiology

Ten-week-old POMC-MAPT-GFP male mice were fed chow or HFD for 4 weeks, and weekly measurements of body weight and food intake were recorded. Animals were sacrificed by decapitation at the beginning of the light cycle, and the fed status was confirmed by blood glucose levels taken before isofluorane anesthesia. The brains were immediately immersed in ice-cold high magnesium artificial cerebrospinal fluid (ACSF) (in mM): NaCl 125, KCl 2.5, NaHCO_3_ 26, NaH_2_PO_4_ 1.25, glucose 11, CaCl_2_ 0.5, MgCl_2_ 2 (pH 7.3, bubbled with 95% O_2_ and 5% CO_2_) to protect from excitotoxicity, and 300 μm acute hypothalamic slices were made using a Leica vibratome. The slices were then recovered for 30 min at 32 °C in normal ACSF containing (in mM): NaCl 125, KCl 2.5, NaHCO_3_ 26, NaH_2_PO_4_ 1.25, glucose 11, CaCl_2_ 2, MgCl_2_ 1 (pH 7.3, bubbled with 95% O_2_ and 5% CO_2_). After recovery, the slices were transferred to HEPES ACSF containing (in mM) NaCl 125, KCl 2.5, NaHCO_3_ 21, HEPES 10, Glucose 11, NaH_2_PO_4_ 1.2, CaCl_2_ 2, MgCl_2_ 2 (pH 7.4). An amount of 50nM CX3CL1 (or vehicle: PBS + 0.1% BSA) or 100 μM Minocycline (or vehicle: water) was added to the HEPES ACSF and the slices were incubated for 2 h at 32 °C. At the end of incubation period, the slices were transferred to a submerged bath in the electrophysiology rig and were perfused with HEPES ACSF with 20nM CX3CL1 or 100 μM Minocycline during recordings. POMC neurons in the arcuate nucleus were visually identified by GFP expression and subjected to whole cell patch clamp using 3–6 MΩ borosilicate glass micropipettes (Sutter Instruments, UK) filled with potassium gluconate internal containing (in mM): K-Gluconate 130, KCl 10, HEPES 10, EGTA 1, Na_2_ATP 2, Mg_2_ATP 2, Na_2_GTP 0.3. All synaptic activity was blocked using DNQX (10 μM), APV (50 μM), and picrotoxin (50μM) in the bath to isolate intrinsic properties of the recorded POMC neurons. Neurons were stimulated with 30 pA current step for variable durations (100–1000 ms) to study their evoked responses to extended depolarizing stimuli. The spontaneous excitatory postsynaptic current (sEPSC) and spontaneous inhibitory postsynaptic current (sIPSC) recordings were made using cesium-based internal solution (in mM) Cs-Me-sulfonate 115, CsCl 20, MgCl_2_ 2.5, HEPES 10, EGTA 0.6, MgATP 4, Na_2_GTP 0.4, Na-phosphocreatine 10, with inhibitory synaptic blockers (Picrotoxin) or excitatory synaptic blockers (DNQX and APV) in the bath, respectively.

Electrophysiology data were acquired with a Multiclamp 700B amplifier and pClamp10 software version 10.7 (Molecular Devices, CA, USA), and sampled at 20 kHz using Digidata. Data were analyzed using Clampfit10 and custom written R scripts.

### 4.6. Tissue Processing

For IHC studies, animals anesthetized using ketamine + xylazine cocktail (140 mg of ketamine and 12 mg of xylazine/kg of body weight) were perfused with ice-cold PBS followed by 4% paraformaldehyde (PFA) in PBS using gravitational flow. Brains were removed, post-fixed in 4% PFA overnight at 4 °C, sucrose (25%) protected, cryosectioned at 30 µm in the coronal plane through the hypothalamus, and stored in freezing solution (phosphate buffer, 30% sucrose, 30% ethylene glycol) at −20 °C for IHC staining. 

For RNA extraction, animals were anesthetized using CO_2_ and rapidly decapitated. The brain was extracted and immediately frozen on dry ice. Hypothalamic blocks were dissected and stored at −80 °C until RNA extraction using RNAeasy kits (Qiagen Cat#74106). 

For microglial cell isolation, immediately following perfusion with PBS, whole brain was dissected and dispersed into a single-cell suspension using Accutase (Innovative Cell Technologies Cat#AM-105) digestion at 4 °C for 30 min. After filtration (100 µm) and centrifugation at 400 g for 10 min, cells were resuspended in FACS buffer and transferred to a 30% Percoll solution. The cell pellet was resuspended and washed twice with FACS buffer. Microglial cells were sorted for CD11b using a FACSAria II Cell sorter (Becton-Dickinson, San Jose, CA, USA). Microglial cells were collected directly into lysis buffer (RNeasy Micro kit, Qiagen Cat#74004) and stored at −80 °C until processing.

### 4.7. Real-Time PCR

Total RNA was extracted using an RNeasy mini kit according to manufacturers instructions (Qiagen Cat#74106) and reverse-transcribed with Multiscribe Reverse Transcriptase (Applied Biosystems). Levels of mRNA for *Pomc, Cx3cr1* and *18S* RNA (internal control) were measured by semiquantitative real-time PCR on an ABI Prism 7900 HT (Applied Biosystems). The primer sequences are as follows: *18S* fwd: 5′-CGG ACA GGA TTG ACA GAT TG-3′, rev: 5′-CAA ATC GCT CCA ACT AA-3′; *Pomc* fwd: 5′-CGC TCC TAC TCC ATG GAG CAC TT-3′, rev: 5′-TCG CCT TCC AGC TCC CTC TTG-3′; *Cx3cr1* fwd: 5′-AGT TCC CTT CCC ATC TGC TC-3′; rev: 5′-GCC ACA ATG TCG CCC AAA TA-3′. The relative expression was calculated by the 2-ΔΔCT method [70].

### 4.8. Immunohistochemical Staining

Free-floating 20 µm coronal brain sections containing the hypothalamus were prepared from mice with hypothalamic injections of AAV–GFP or AAV–CX3CL1-s using a cryostat (Leica, Deer Park, IL, USA). The sections were washed with PBS, blocked with 5% normal donkey serum (Jackson Immuno Research Laboratories Cat#017-000-121) and incubated overnight with rabbit anti-HA (1:1000; Cell Signaling C29F4) or chicken anti-GFP (1:5000; Abcam Cat#13970). Hypothalamic sections from Cx3cr1-GFP mice were incubated with either rabbit anti-Iba1 (1:1000; Wako Cat#019-19741), mouse monoclonal Anti-GFAP-Cy3 (1:1000; Millipore Sigma Cat#C9205), mouse anti-NeuN (1:1000; Millipore Sigma Cat#MAB377) or chicken anti-Vimentin (1:5000; Abcam Cat#ab24525). After primary incubation overnight at 4 °C, all wells were processed for 1h with appropriate fluorescent secondary antibodies (Alexa Fluor 594 donkey anti-rabbit (1:1500, Life Technologies Cat#R37119); Alexa Fluor 488 donkey anti-chicken (1:500, Jackson Immuno Research Cat#703-545-155); Alexa Fluor 594 goat anti-rabbit (1:500, Invitrogen); Alexa Fluor 594 goat anti-chicken (1:500, Jackson Immuno Research Cat#103-585-155); Alexa Fluor 594 donkey anti-mouse (1:500, Invitrogen Cat#R37115). Stained sections were imaged using a cooled CCD camera attached to a DS-Ri1 epifluorescence microscope (Nikon, Melville, NY, USA). 

### 4.9. Statistical Analyses

All results are presented as mean ± SEM. Statistical analysis using Prism (GraphPad) involved unpaired and paired two-tailed Student’s *t* tests for data with normal distribution, and Kolmogorov–Smirnov test (KS test) for data with non-normal distribution. Two-way ANOVA with post-hoc Tukey’s multiple comparisons testing was used where noted. Probability (*p*) values of less than 0.05 were considered statistically significant.

## Figures and Tables

**Figure 1 ijms-23-06380-f001:**
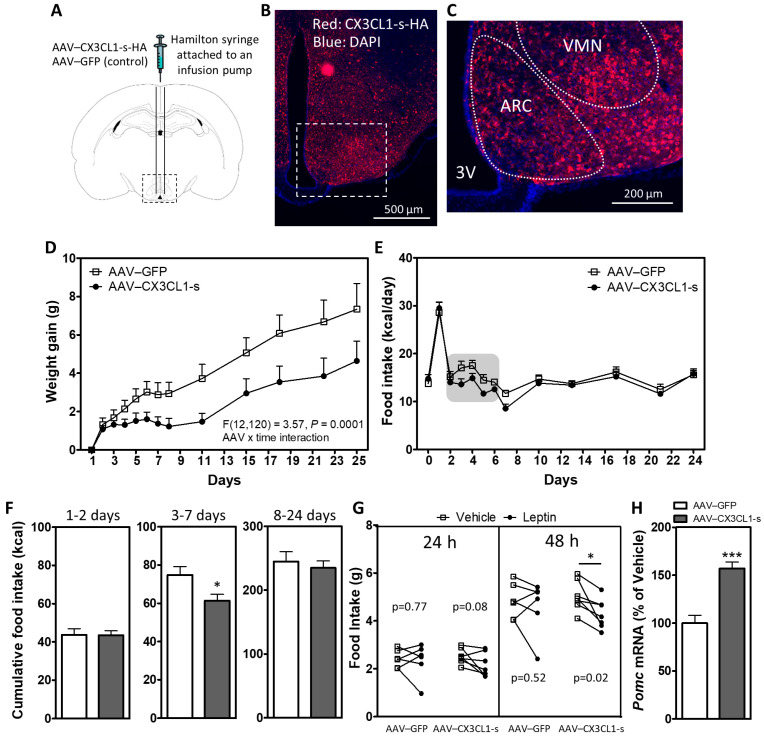
Hypothalamic overexpression of soluble CX3CL1 reduces weight gain, food intake and improves leptin sensitivity in DIO mice. (**A**) Schematic diagram of a mouse coronal brain section indicating bilateral administration of AAVs into the mediobasal hypothalamus. (**B**,**C**) Representative image of a hypothalamic section from a mouse treated with AAV–CX3CL1-s. CX3CL1-s overexpression in the MBH is indicated by HA staining (red). Higher magnification view in C shows broad expression in the arcuate nucleus (ARC) and ventromedial hypothalamus (VMH). (**D**,**E**) Body weight gain (**D**) and 24 h food intake (**E**) in HFD-fed mice with hypothalamic overexpression of GFP (AAV–GFP) or the soluble isoform of CX3CL1 (AAV–CX3CL1-s) measured daily over the first week and then bi-weekly until day 25 of HFD. (**F**) Cumulative food intake over days 1–2 (left panel), 3–7 (middle panel) and 8–24 (right panel). (**G**) 24 h and 48 h food intake after i.p. saline or leptin injections (2 µg/g/day) in AAV–GFP and AAV–CX3CL1-s mice on HFD. Data are presented as mean ± SEM of at least 6 animals per group. (**H**) Hypothalamic mRNA expression of *Pomc* in chow-fed AAV–GFP and AAV–CX3CL1-s mice. Data are expressed as a percentage of AAV–GFP control; n = 6 per group. * *p* < 0.05; *** *p* < 0.001.

**Figure 2 ijms-23-06380-f002:**
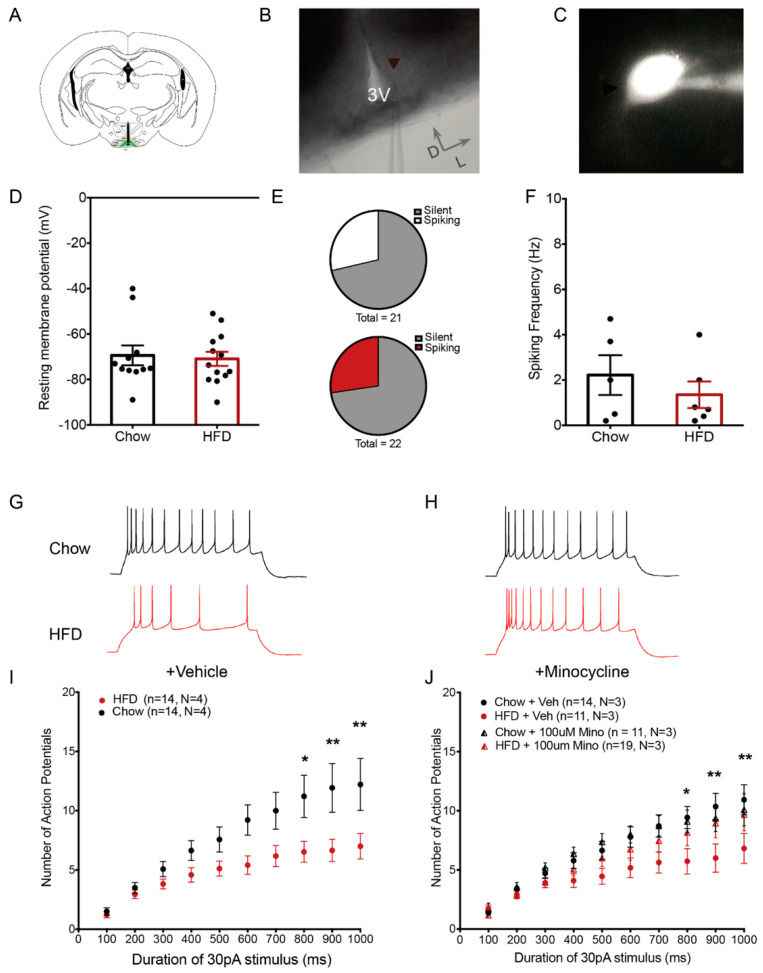
Minocycline restores intrinsic excitability of POMC neurons in HFD-fed mice. (**A**) Schematic diagram showing a coronal hypothalamic slice. The arcuate nucleus (ARC), where GFP-expressing POMC neurons were visualized and recorded from, is highlighted in green. (**B**) 4X DIC image showing the ARC region and recording pipette attached to a recorded POMC neuron. (**C**) Representative fluorescent image of a GFP-expressing POMC neuron with attached recording pipette filled with internal solution containing fluorophore Alexa-594. (**D**) Resting membrane potential of recorded POMC neurons (Chow: −69.38 ± 4.3 mV, n = 11, N = 3; HFD: −70.92 ± 3.1 mV, n = 13, N = 3). (**E**) Proportion of recorded POMC neurons showing spontaneous activity (upper: Chow (6 out of 21 neurons), lower: HFD (6 out of 22 neurons)). (**F**) Frequency of action potentials in spontaneously firing POMC neurons (Chow: 2.220 ± 0.87 Hz, n = 6, N = 5; HFD: 1.35 ± 0.58 Hz, n = 6, N = 5). (**G**,**H**) Representative traces of POMC neuron responses after stimulation by 1000 ms 30 pA current step in brain slices from chow and HFD-fed mice treated with vehicle (**G**) or minocycline (100µM) (**H**). (**I**,**J**) Population data of evoked responses of POMC neurons to variable duration of stimulus in brain slices from chow and HFD-fed mice incubated with vehicle (**I**) or minocycline (**J**). Lowercase (n) indicates number of neurons and uppercase (N) number of animals. * *p* < 0.05; ** *p* < 0.01.

**Figure 3 ijms-23-06380-f003:**
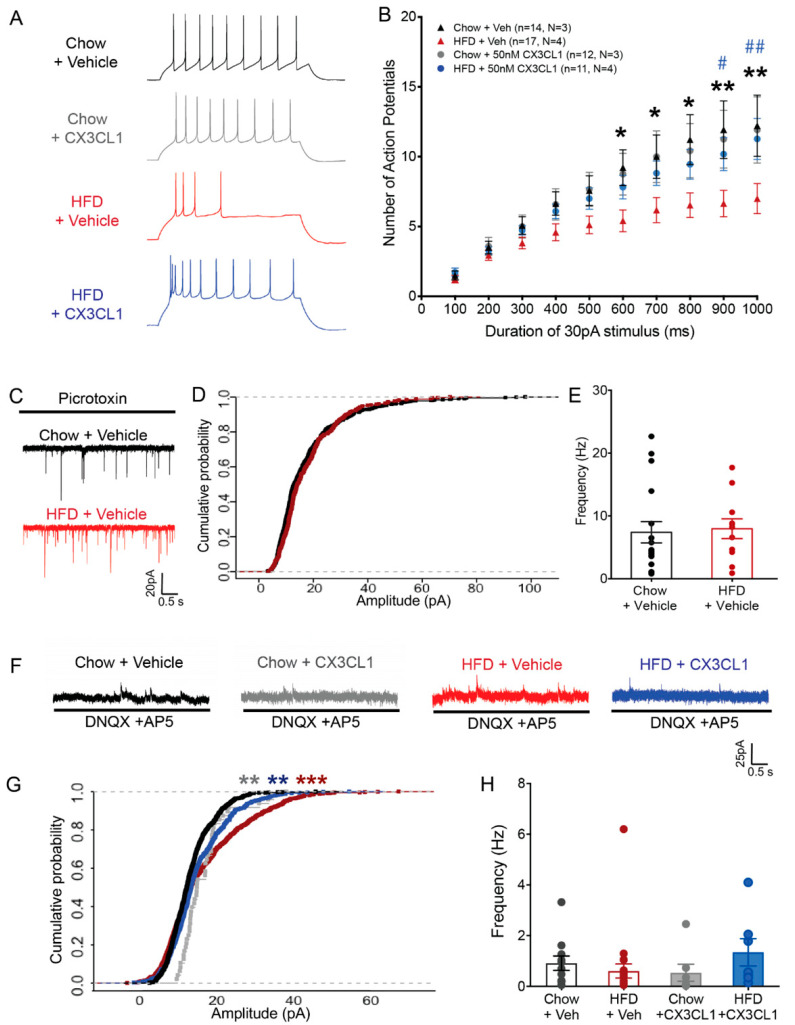
CX3CL1 restores intrinsic excitability of POMC neurons in DIO mice. (**A**) Representative traces of POMC neurons after stimulation by 1000 ms 30 pA current step (top: stimulus step; bottom: responses: black: Chow + Veh, gray: Chow + CX3CL1, red: HFD + Veh, blue: HFD + CX3CL1). (**B**) Population data of evoked responses of POMC neurons to variable duration of stimulus after vehicle or CX3CL1 treatment in brain slices from chow and HFD-fed mice (* denotes *p* values for Chow + Veh and HFD + veh comparison, #, ## denotes *p* values for HFD + veh and HFD + CX3CL1 comparison). (**C**–**E**) Representative traces (**C**), cumulative probability plot of amplitude (**D**), and frequency plot (**E**) of sEPSCs for POMC neurons recorded in chow and HFD brain slices (Frequency of sEPSCs: Chow: 7.41 ± 1.69 Hz, n = 17, N = 3; HFD: 7.96 ± 1.56 Hz, n = 11, N = 3). (**F**–**H**) Representative traces (**F**), cumulative probability plot of amplitude (**G**), and frequency plot (**H**) of sIPSCs for POMC neurons recorded in chow and HFD brain slices incubated for 2 h with vehicle or CX3CL1 (Frequency of sIPSCs: Chow + Veh: 0.76 ± 0.28 Hz, n = 18, N = 4; HFD + Veh: 0.61 ± 0.28 Hz, n = 22, N = 4; Chow + CX3CL1 = 0.54 ± 0.33 Hz, n = 7, N = 2; HFD + CX3CL1: 1.3 ± 0.54 Hz, n = 7, N = 2). Lowercase (n) indicates number of neurons and uppercase (N) number of animals. * *p* < 0.05; ** *p* < 0.01; *** *p* < 0.001.

**Figure 4 ijms-23-06380-f004:**
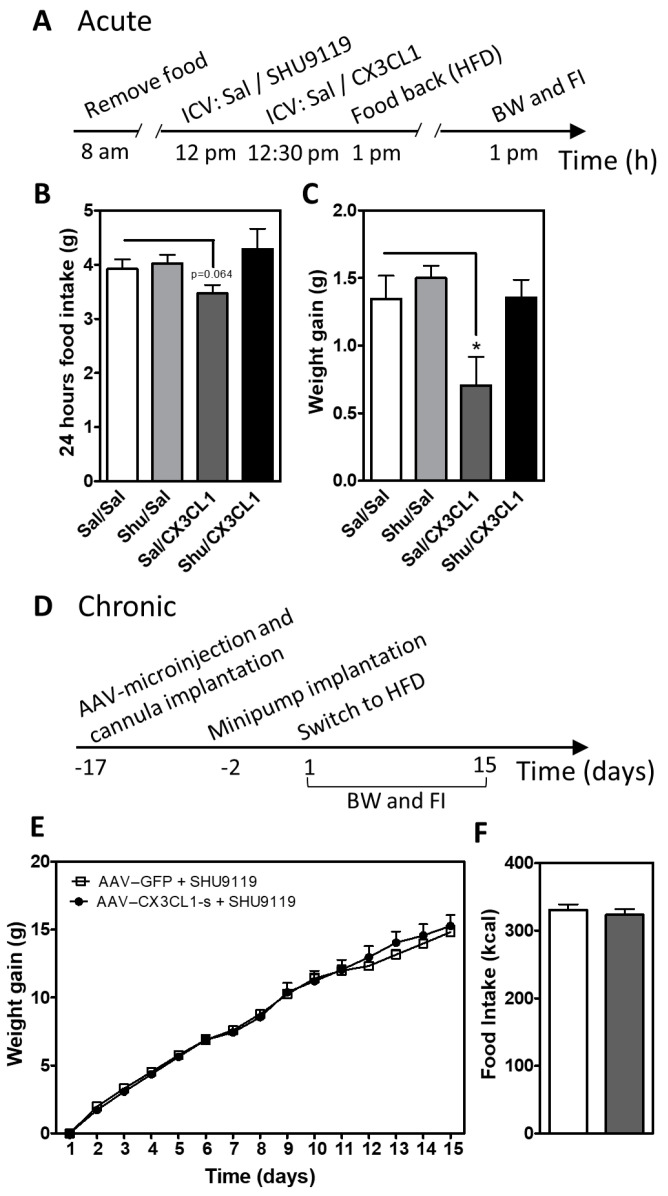
Central CX3CL1 does not reduce weight gain and food intake in mice treated with MC3R/MC4R antagonist**.** (**A**) Time scale of the experimental approach used for acute i.c.v. administration of SHU9119 and CX3CL1. (**B**,**C**) Average of cumulative 24 h food intake (**A**) and body weight gain (**B**) in mice treated with saline (Sal) or SHU9119 i.c.v. and 30 min later with Sal or CX3CL1 i.c.v. Data are presented as mean ± SEM of at least 5 animals per group. * *p* < 0.05. (**D**) Time scale for chronic i.c.v. administration of SHU9119 in mice previously subjected to hypothalamic viral microinjection of AAV–GFP and AAV–CX3CL1. (**E**,**F**) Body weight gain (**E**) and total food intake (**F**) measured in mice with hypothalamic overexpression of CX3CL1-s or GFP control that received SHU9119 i.c.v. infusion over 2 weeks (compare with Figure 1D). Data are presented as mean ± SEM of 10 animals per group. * *p* < 0.05.

## Data Availability

Not applicable.

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
