# Peer review of "CX3CL1 Action on Microglia Protects from Diet-Induced Obesity by Restoring POMC Neuronal Excitability and Melanocortin System Activity Impaired by High-Fat Diet Feeding"

_ijms, 2022, doi:10.3390/ijms23126380_

Round 1
Reviewer 1 Report
The manuscript is well-written and very interesting. The author's have elaborated the role of CX3CL1 reducing the weight gain when overexpressed in the MBH, reporting an interesting interplay between microglial activation and POMC neural dysfunction in mice.
I have only some issues in regard to the method section:
- please describe if the mice during the surgical procedure has been anesthesized (with what compounds and what conc.)? If yes (like mentioned in Tissue processing), then please discuss the role of the usage of anesthetics altering the metabolic profile in mice;
- please give a detailed description of Leptin sensitivity method;
- for each device/machine/equipment state the manufacturer and state;
- for any product used please state the product cat. no., manucfaturer and country;
- what housekeeping genes did you use for the RT-PCR experiments (18S?)? If yes, why this gene and not any other well-known genes like GAPDH, HMBS, etc? How did you express the relative expression of your genes of interest (Schmittgen method or?)?
- (curious about your findings presented in Suppl. Fig. 1) what about the CX3CR1 expression in oligodendrocytes?
Author Response
Reviewer 1
Comments and Suggestions for Authors
The manuscript is well-written and very interesting. The author's have elaborated the role of CX3CL1 reducing the weight gain when overexpressed in the MBH, reporting an interesting interplay between microglial activation and POMC neural dysfunction in mice.
We thank the reviewer for the kind remarks.
I have only some issues in regard to the method section:
- please describe if the mice during the surgical procedure has been anesthesized (with what compounds and what conc.)? If yes (like mentioned in Tissue processing), then please discuss the role of the usage of anesthetics altering the metabolic profile in mice;
All surgical procedures (viral injection and cannula implantation) were performed under isoflurane anesthesia. However, all in vivo studies including body weight, food intake, and leptin sensitivity were performed at least 10 days after surgery in awake and free moving mice. We have now clarified this important point in the method section.
- please give a detailed description of Leptin sensitivity method;
Thanks. We have included a more detailed description in this section.
- for each device/machine/equipment state the manufacturer and state;
Manufacture information has been added.
- for any product used please state the product cat. no., manucfaturer and country;
Catalog numbers are now included in methods.
- what housekeeping genes did you use for the RT-PCR experiments (18S?)? If yes, why this gene and not any other well-known genes like GAPDH, HMBS, etc? How did you express the relative expression of your genes of interest (Schmittgen method or?)?
18S rRNA was used as a housekeeping gene (internal control) in RT-PCR. It has been described that 18s shows less variance in expression across a variety of treatment conditions than β-actin and GAPDH. The 2-ΔΔCT method was used for relative gene expression. We have included this information in the method section.
- (curious about your findings presented in Suppl. Fig. 1) what about the CX3CR1 expression in oligodendrocytes?
We did not assess the expression of Cx3cr1 in oligodendrocytes.
Reviewer 2 Report
Very interesting and methodically rich paper, results presented and discussed in detail. As it provides further evidence of the close cooperation between the immune and nervous systems this time in the context of the mechanisms of obesity.The topic is adequately supported by citations, thoroughly discussed, and the results described in great detail.
I just suggest you to add to the key words – fractalkine
Author Response
Reviewer 2
Comments and Suggestions for Authors
Very interesting and methodically rich paper, results presented and discussed in detail.
I just suggest you to add to the key words – fractalkine
We thank the reviewer for the kind remarks and completely agree in adding fractalkine as a key word.
Round 2
Reviewer 1 Report
Dear author's thanks for the rapid revision and upgraded version of your manuscript. I'm satisfied with the newly version and recommend the acceptance of your manuscript.